# Large Scale Vat-Photopolymerization of Investment Casting Master Patterns: The Total Solution

**DOI:** 10.3390/polym14214593

**Published:** 2022-10-29

**Authors:** Farzaneh Sameni, Basar Ozkan, Sarah Karmel, Daniel S. Engstrøm, Ehsan Sabet

**Affiliations:** 1Wolfson School of Mechanical, Electrical and Manufacturing Engineering, Loughborough University, Loughborough LE11 3TU, UK; 2Additive Manufacturing Centre of Excellence Ltd., 33 Shaftesbury Street South, Derby DE23 8YH, UK; 3Polyfos Ltd., London SW11 6NF, UK

**Keywords:** vat photopolymerization, investment casting, casting pattern, thermoplastic-thermoset photopolymers, wax-filled casting resin

## Abstract

The material properties and processing of investment casting patterns manufactured using conventional wax injection Molding and those manufactured by vat photopolymerization can be substantially different in terms of thermal expansion and mechanical properties, which can generate problems with dimensional accuracy and stability before and during ceramic shelling and shell failures during the burn-out of the 3D printed casting patterns. In this paper and for the first time, the monofunctional Acryloyl morpholine monomer was used for 3D printing of casting patterns, due to its thermoplastic-like behavior, e.g., softening by heat. However, the hydrophilic behavior of this polymer led to an incorporation of up to 60 wt% of Hexanediol diacrylate, to control the water absorption of the network, which to some extent, compromised the softening feature of Acryloyl morpholine. Addition of a powdered wax filler resulted in a delayed thermal decomposition of the polymer network, however, it helped to reduce the thermal expansion of the parts. The dimensional accuracy and stability of the wax-filled formulation indicated an excellent dimensional tolerance of less than ±130 µm. Finally, the 3D printed patterns successfully went through a burn out process with no damages to the ceramic shell.

## 1. Introduction

Casting, or pouring molten metal into molds with negative cavity of final parts, is one of the most original methods of manufacturing metallic parts. Among various major casting techniques (e.g., sand casting, die casting, shape casting, etc.), investment casting (IC) is one that produces parts with an excellent dimensional accuracy and surface finish, which is hard to manufacture with any other casting method. IC is capable of yielding tolerances in the range of ±1% of the nominal dimensions and a roughness of 3.2 µm [1]. Investment casting or lost wax casting ages back to 1500 BC and today it is used to manufacture complex geometries such as turbine blades with intricate cooling channels [2,3]. Investment casting process starts with manufacturing a wax pattern of the final product using an injection mold. The wax pattern is then dipped or “invested” into ceramic slurry, and covered by refractory ceramic powder, these two steps are repeated until the desired “ceramic shell” thickness is achieved. After this the wax pattern is either melted or burnt out of the ceramic shell. The ceramic shell is then cast, followed by knocking down the shell to obtain the metal product [2,4,5,6]. The excessive difficulty of machining Nickle super alloys, used for manufacturing gas turbine blades, makes investment casting the first resource of manufacturing gas turbine blades [1,2,3,5,7,8]. However, due to the complex shape of hot-section turbine components, machining of the injection mold used for wax pattern production is often extremely time consuming and costly [3,5,6,7,9,10]. Hence, using additive manufacturing methods to directly produce the casting patterns offers several advantages including cost and production time reduction, regardless of complexity of the components as no mold is needed in this process [3,6,7,11].

Casting patterns used in investment casting process needs to fulfil certain criteria, including low ash content after burnout, dimensional precision and stability, enough handling/assembling strength, and high surface quality and wettability (for a successful ceramic shelling in the investment casting process) [5,12].

Additive manufacturing (AM) or 3D printing, is a manufacturing method that produces the parts directly from a CAD model and in a layer-by-layer manner [3,5,13]. Among other AM methods, vat photopolymerization (VPP) was the first AM technique to be invented and commercialized in 1980s under the name “stereolithography (SLA)”. The feed stock material in VPP 3D printing is a photosensitive resin (photoresin) that selectively cures and solidifies by light illumination to create the final part [13,14,15]. Compared to other AM techniques, VPP often offers a larger build area, and produces parts with better surface finish and dimensional accuracy [13,15]. Therefore, using VPP to produce casting patterns with high precision and good surface qualities gained interest not long after it was invented, with one of the first investigations conducted by Jacobs in 1992 [16].

The material used in vat photopolymerization techniques is a photosensitive resin that solidifies upon exposure to light. A photosensitive resin is a combination of different components including monomers and oligomers, reactive diluents, photoinitiator, light absorbers, and colorants and pigments. After irradiation with light, the photoinitiator species in the photoresin mixture turn into reactive free radicals that activate the monomers and oligomers and start the polymerization (curing) process [3,13,14,17]. (Meth)acrylate monomers are commonly used in photoresin systems. To achieve a suitable physical and mechanical properties. A combination of di- and multi-functional (meth)acrylate monomers are used to form crosslinked networks [17]. As a result of this cross-linked network, photoresins, known as thermosetting polymers, show different behavior compared to thermoplastic materials, e.g., wax. In contrast with thermoplastics, thermosetting materials do not melt when heated, instead they tend to burn and degrade. The crosslinked nature of thermosetting materials results in high strength, stiffness, and hardness compared to thermoplastic polymers [3,18,19,20]. This is one of the substantial differences between the traditional casting patterns obtained by injection molding of refractory wax, and the novel casting patterns made by vat photopolymerization 3D printing techniques. As an example, in contrast with traditional wax patterns, VPP 3D printed casting patterns cannot be melted out of the ceramic shell in industrial autoclaves [3,5], instead they need to be directly burnt out in firing furnaces [21].

Hollowed patterns, made by 3D printing, save material, and reduce the overall thermal expansion of the parts during burnout process. Attempts to develop new thermosets with low thermal expansions, to be used in VPP of casting patterns, was made back in early 1990s, and continued since [22,23,24,25]. Quickcast^®^, as one of the pioneer commercial software packages, was made to hollow the casting patterns to enhance the survival chance of ceramic shells during burnout. This internal structure intended to induce inwards buckling of the patterns during the burnout process [22,24]. More internal structures were developed by other researchers with similar intentions [26,27,28]. Although 3D printing hollow casting patterns with internal structures is a prerequisite for simple and bulky components, it is not enough for parts with very thin sections that cannot be printed hollow in practice.

Studies on the use of VPP to produce casting patterns started in early 1990s and continued until around 2012, but there are very little research articles reporting further investigations on 3D printed casting patterns for investment casting application thereafter [29]. Since no commercial product has since been introduced to the market for general use in the investment casting of large and complex components, except for small patterns in jewelry and dental applications, it can be arguably concluded that the research on materials for such applications have not been fully successful, so far.

In a work conducted by Carneiro et al. [30], the casting pattern for a metallic cellular component was prepared using PLA and fused filament fabrication technique. In this work, the ceramic mold was created by infiltrating a plaster material through the cellular casting pattern, followed by a thermal cycle to cure the mold and then burnout of the pattern. They reported an overall deformation of up to 10% caused by their 3D printing technique, which can be significant for aero and industrial turbine components, addressed in this paper. In another research [31], the use of casting patterns manufactured by selective laser sintering (SLS) (using PrimeCast^®^) and binder jetting (using PMMA) and their cast parts using aluminum alloy A356 was investigated. The result of their micro-CT measurements revealed a dimensional deviation of ±0.7 to ±1.5 mm, which exceeds the acceptable tolerances for net-shape casting of many engineering components. Andrew et al. [32] Used a wax filament in a desktop 3D printer to fabricate casting patterns. However, apart from the poor quality of extrusion based 3D printing techniques, the authors suggested that casting patterns made by this method are not suitable for applications where dimensional accuracy matters. Wang et al. [33], who presented one of the most recent studies in AM-made casting patterns, compared the performance of VPP and SLS manufactured casting patterns and concluded that patterns fabricated by VPP process may cause shell cracking and are not suitable for the thin-walled and complex geometries. Recently, Badanova et al. [34] systematically investigated the impact of different print parameters of the VPP 3D printed casting patterns on the dimensional accuracy and geometrical features of casting patterns and their cast products. They reported layer thickness and build angle as the most impactful parameters in the dimensional quality of the casting patterns. The same authors also reported the dimensional sensitivity to the geometrical feature of the pattern and their size. Although there are some other reports on the use of 3D printed patterns in investment casting [35,36,37], to our knowledge there is no report on VPP 3D printing of casting patterns for the hot section components of aero and gas turbines, where the casting products of such patterns must satisfy the needs for high geometrical complexity, large sizes, and very tight dimensional tolerances. Therefore, the focus of this research is to develop a photoresin formulation for VPP 3D printing of casting patterns of such extreme applications, where change of design can have significant impact on development cost and time.

The castable photoresin must offer a minimum thermal expansion and low mechanical properties at high temperatures, while displaying sufficient mechanical properties and dimensional accuracy at room temperature for the ease of handling and dimensional stability of the casting pattern before ceramic-shelling.

Use of thermoplastic and soluble photocurable monomers to produce 3D printed sacrificial molds is suggested by Deng et al. [38]. Isobornyl acrylate and 4-acryloylmorpholine (ACMO) as one of such material, is studied in their work. These monomers are a group of monofunctional monomers in which the chain only grows in one direction and the typical covalence crosslinking bonds between the chains are replaced with hydrogen bonds that account for solubility, softening, and melting of the cured polymer [38,39,40].

To date, 3D printing of thermoplastic materials has been dominated by the filament- or powder-based AM techniques. 3D printing thermoplastic parts using vat photopolymerization techniques is a novel area that only a few researchers have recently discussed in their works [38,39]. To the best of our knowledge, the current literature is lacking a wholesome solution when it comes to additively manufactured casting patterns that not only survive the burnout process, but also present high precision for applications with tight dimensional tolerances. This paper, for the first time, presents the use of thermoplastic monomers for VPP 3D printing of sacrificial casting patterns of large, complex, and highly accurate components and investigates the impact of adding a wax filler to photocurable formulations as a popular approach in formulation photocurable resins for this application.

## 2. Materials and Methods

### 2.1. Materials and Mixture Preparation

#### 2.1.1. Unfilled Photoresins

To prepare the photoresins for this study, two photocurable monomers, Acryloyl morpholine (ACMO) (Rahn AG, Zürich, Switzerland) and Hexanediol diacrylate (HDDA) (Miwon, Anyang, South Korea) were used as the base monomers. Due to the hydrophilicity of ACMO polymer network [41], ACMO and HDDA were mixed in 100:0, 80:20, 60:40, and 40:60 ratios, respectively, to control the water absorption and swelling of the final polymer network. The photoinitiator, Phenylbis (2,4,6-trimethylbenzoyl) phosphine oxide (BAPO) (Rahn AG, Zürich, Switzerland), was added on top each photoresin mixture in 1.5 wt% of the total monomers weight. The mixtures are labelled as A100, A80, A60, and A40 based on their ACMO content, respectively (Table 1). Figure 1 presents the chemical structures of the monomers and the photoinitiator used in this work. For a homogenous distribution of monomers and the complete dissolution of the photoinitiator, photoresins were mixed in a planetary high-speed mixer (SpeedMixer DAC 150 FVZ, Hauschild, Germany) for 15 min at 20,000 rpm. Due to the difficulties in handling of A100 caused by its excessive tackiness and adhesion to the build platform, this mixture was eliminated from the experiments. A series of tests including Degree of Conversion (DC) measurement, Viscosity measurements, Tensile testing, and Water absorption were conducted using these mixtures. Tensile strength and the water absorption of the unfilled mixtures were used to narrow down the monomer combinations for the wax-filled studies.

#### 2.1.2. Wax-Filled Photoresins

Oxidized polyethylene wax (Shamrock Technologies, Tongeren, Belgium) was used in one of the formulations in 40, 50, and 60 Vol% to obtain the optimum wax addition to the mixtures. Photoresins were mixed for 15 min in the planetary mixture to ensure the homogenous distribution of wax particles. Based on the viscosity measurements and the flowability of the mixtures, as the critical characteristics for successful printing, the optimal wax content was selected. This was then added to all photoresin mixtures. Table 1 presents the labelling and contents of each mixture. Wax-filled mixtures were also assessed for their DC, tensile properties, water absorption, as well as thermal analysis tests and dimensional investigations.

The impact of a light-absorbing pigment on the wax-filled photoresins was also investigated. To do this, 10 g of carbon black powder (Monarch 1300) (Cabot, Aloharetta, GA, USA) was diluted in 15 g of HDDA. This was added in 1.5 wt% on top of the total monomers weight to the shortlisted mixture.

### 2.2. Viscosity Measurements

The viscosity measurements were taken using a rheometer device (Brookfield RST, Middleboro, MA, USA). Tests were conducted by loading less than 2.0 mL of sample on the temperature-controlled static holding plate. The gap size was set to 100 µm to the opposite stainless-steel parallel plate with a diameter of 40 mm. The viscosity was measured at room temperature, for 120 s, and: (1) Under constant shear rate of 500 s^−1^ for unfilled photoresins and (2) Under constant shear rate of 100 s^−1^ for wax filled photoresins. Since the unfilled photoresins had a very low viscosity, the rheometer device was unable to measure their viscosities under low shear rates, e.g., 100 s^−1^. Therefore, the shear rate of 500 s^−1^ was applied to this group of photoresins.

### 2.3. Degree of Conversion

To measure the DC of the samples, the Attenuated Total Reflection (ATR) module of a FTIR spectrometer (Bruker Alpha, Billerica, MA, USA) was used. The spectrometry was conducted with 40 scans and a resolution of 2 cm^−1^. Test samples were prepared by curing the photoresins using a Sonic Mega 8K 3D printer to a thickness of 250 µm. The excess uncured resin was wiped clean from the top of the samples. The spectrums were taken from the surface in contact with the screen during curing. The uncured photoresin of each test sample was used as the reference and the DC was calculated using Equation (1) [42]. In this equation A_C=C_ is the area of the peak at 1635 cm^−1^ for C=C double bonds, and A_C=O_ is the area of the peak at 1720 cm^−1^ as the internal standard for DC calculations, in both uncured photoresin (M) and cured polymers (P).
(1)$$\mathrm{DC}=1-\left[\frac{{\left(\frac{A_{C=C}}{A_{C=O}}\right)}_{P}}{{\left(\frac{A_{C=C}}{A_{C=O}}\right)}_{M}}\right]\times 100$$

DC of A80, A60, A40, and the 50 vol% wax filled counterparts, A80W50, A60W50, and A40W50, were measured using the abovementioned method. The effect of post-curing and black pigment addition were also investigated on A40W50 as the shortlisted photoresin based on later-mentioned experimental results.

### 2.4. Tensile Strength

Six tensile specimens according to ASTM D638 were prepared for A80, A60, A40, and their 50 vol% wax filled counterparts, A80W50, A60W50, and A40W50. Tests were conducted on a (MultiTest 2.5-dv, Mecmesin, Horsham, UK) mechanical testing machine with 1 mm/min test speed. Test samples were printed vertically in Z direction with a layer thickness of 100 µm on a screen-based vat photopolymerization 3D printer (Sonic Mega 8k, Phrozen, Hsinchu, Taiwan).

### 2.5. Water Absorption

To measure the water absorption of the mixtures, three disk-shaped samples (3.2 × 50.8 mm) were printed according to ASTM D570-98. To examine the 24-h water absorption of samples, they were first fully dried for 24 h at 50 °C and then immersed in distilled water for 24 h at 23 °C. Samples were weighed in their dry and wet states and the percentage of water absorption was calculated using Equation (2).
(2)$$\mathrm{Water}\;\mathrm{bsorption}\%=\frac{\mathrm{Wet}\;\mathrm{Weight}-\mathrm{Dry}\;\mathrm{Weight}}{\mathrm{Dry}\;\mathrm{Weight}}\times 100$$

### 2.6. Thermal Analysis

Thermal analysis tests, including Thermogravimetric Analysis and Thermomechanical Analysis (TGA and TMA) were performed. TGA (SDT Q600, TA Instruments, New Castle, DE, USA) was used to understand the thermal degradation regime of the printed formulations (during burnout) and their residual ash content after burnout process. The tests were carried out from room temperature to 800 °C with a heating rate of 20 °C/min. To measure the thermal expansion of the printed pats, 5 × 5 × 5 mm cubic samples were tested from room temperature to 150 °C with a heating rate of 5 °C/min in a TMA machine (TMA Q400, TA Instruments, New Castle, DE, USA).

### 2.7. Deflection Test for Burnout Softening Investigation

The effect of material and wall thickness of the parts on the deflection and softening behavior of the parts were examined on the shortlisted photoresins (A40, A40W50, and A40W50-PC). To do this, test-strips of 20 × 60 mm with 0.8 mm, 1.3 mm, and 1.8 mm thicknesses were prepared and heated to 100 °C, 150 °C, and 200 °C and dwelled for 15 min while they were vertically fixed in a clamp at one end. The deflection of the test strips at different temperature were visually inspected as an indication of the softening behavior of the investment casting patterns during the burnout process and prior to decomposition of the polymers.

### 2.8. Dimensional Measurements

The dimensional accuracy and stability of a typical solid IGT blade (Appendix A) were assessed using a 3D scanner machine with an accuracy of 5 µm (Altra 120, Nikon Metrology, Tring, UK). The scanned point clouds were analyzed using Focus Inspection Software (Focus 11.2, Nikon Metrology, Tring, UK) with the Global CAD Compare feature. Two casting patterns of the abovementioned blade were printed with a wall thickness of 0.8 mm on a Phrozen Sonic Mega 8K 3D printer with a working wavelength of 405 nm. The blades were then scanned shortly after the printing. After the initial scanning, one blade went through the post-curing process and then was rescanned, to study the effect of post-curing on its dimensional accuracy. The post-curing was conducted in an UV oven (Cure L, Photocentric Ltd., Peterborough, UK) at 60 °C for 15 min. Both post-cured and not post-cured parts were rescanned after 7, 14, and 28 days of storage in ambient condition to evaluate their dimensional stability.

### 2.9. Burnout of Casting Pattern

The burnout performance was examined using a more complex and larger solid blade geometry (Appendix A). Preformulated ceramic slurry (Remasol^®^ JUS-DIP™, Remet, Kent, UK) and refractory coatings (Remasil 50 30/60 and Remasil 50 16/30, Remet, Kent, UK) were used to form the ceramic shell on the casting patterns. Remasil 50 30/60 was applied as the refractory coating for the first two coating layers to create a smooth surface roughness on the surface of the casting pattern, and Remasil 50 16/30 was used for the subsequent coatings. A total of nine slurry and stucco coatings were applied on the casting pattern which resulted in a shell thickness of approximately 10 mm. The shelling was completed with a final layer of slurry (no stucco coating) for smooth outer layer. After the ceramic shelling process was completed, the casting pattern went through the burnout process. The burnout was performed in an electric furnace (Enitherm THA250F, Rohde, Hanau, Germany). The furnace was set to reach to 450 °C (in 15 min), with a dwelling time of 60 min, and then to reach 800 °C in 15 min.

## 3. Results and Discussion

### 3.1. Viscosity

The results of viscosity measurements for unfilled and wax-filled photoresins are illustrated in Figure 2. The viscosity of the unfilled photoresins showed to be very close to each other and extremely low (less than 0.02 Pa.s). Hence the impact of wax on the viscosity of photoresins was only studied on one of the mixtures (A40). Addition of wax in 40 vol%, 50 vol%, and 60 vol% resulted in a significant increase in viscosity. Although the maximum measured viscosity of 60 Vol% wax filler falls into the printable viscosity range for photoresins suggested by the literature [38,43], its poor flowability under no shear (gel-like appearance when no external shear force is applied) (Figure 2c) led to exclusion of 60 Vol% wax containing mixtures from the rest of this research, as most commercial VPP 3D printers have no dispensing systems to replenish the material after each printed layer. Between the remaining wax contents, both 40 Vol% and 50 Vol% showed printable characteristics, therefore, 50 Vol% (as the highest printable wax filled photoresin) was taken forward for further investigations.

### 3.2. Degree of Conversion (DC)

Table 2 presents the DC of unfilled and 50 Vol% wax filled photoresins. The measurements indicate that increasing the content of HDDA from 20% to 60% reduces the monomer conversion in photoresins. This is attributed to the covalence cross-linking of HDDA in the polymer network which results in gelation and vitrification at lower conversions after which the polymerization slows down [44]. Addition of wax has caused a reduced DC in all mixtures as wax particles can block the progress of monomer conversion process. This can be attributed to the blocking of the reactive polymer sites by unreactive wax particles, and thus a lower monomer conversion.

### 3.3. Tensile Tests

Although increasing the HDDA content from 20% to 60% reduces the DC of cured photoresins, the tensile strength of both filled and unfilled photoresins improves by increasing the HDDA content (Table 3, Figure 3). This is explained by HDDA’s bifunctionality that enables the covalent crosslinking of the polymer network as opposed to the hydrogen bonds provided by ACMO monomers. Wax particles dispersed in photoresins act as porosities within the polymer matrix and disturb the integrity of the polymer network, hence the tensile strength of all mixtures reduces after addition of wax. In addition, wax particles reduce the degree of double bond conversion and thus also of crosslinking, resulting in reduced ultimate tensile strength (UTS) and Young’s Modulus. Table 2 shows that despite the increased DC in A40W50-B-PC compared to its un-post-cured counterpart (A40W50-B), its UTS has remained the same. To explain this, the differences in principles of ATR and mechanical testing, as well as the role of Black pigment should be considered. The presence of Black pigment in the sample limits the UV light penetration in the parts (e.g., during post-curing). While ATR only analyses a few microns of the surface of the specimen [45], mechanical testing examines the entire bulk of the test sample. Therefore, by using ATR the DC of the surface of the samples are measured which are freely exposed to UV light during post-curing, whereas tensile testing shows the property of the body of the samples, in which the sub-surface does not received UV exposure during post-curing, due to light blockage by the black pigment content. Therefore, although the surface of the pigmented samples is post-cured, the bulk retains their un-post-cured (or so called green) properties.

### 3.4. Water Absorption

A 24-h water absorption test was carried out according to ASTM D570-98. Table 4 concludes the average water absorption of each mixture. As mentioned in Section 2.1.1, ACMO monomers form a hydrophilic polymer network and are extremely prone to humidity and therefore the polymer network is water soluble. Hence, the ACMO content of the test samples has a direct impact on their water absorption. An increased water absorption leads to excessive swelling, deformation, and softening of the parts stored in ambient, which makes them unsuitable for the intended application. The changes in the appearance of the samples after a 24-h immersion in distilled water are shown in Figure 4. In contrast with A80W50 samples which contain the highest amount of ACMO, A40W50 samples with less ACMO content show no visible signs of water absorption, and less than 1 wt% weight increase. Therefore, copolymerizing ACMO with higher content of HDDA as a strategy to reduce the water absorption of the polymer network appears promising. In the wax containing samples, the impact of ACMO content on water absorption is less significant in general. As the content of ACMO drops, adding wax to the formulation can even more contain the water absorption. This is due to the hydrophobicity of the wax particles which reduces water absorption by the polymer network.

As A40 and A40W50 showed the highest tensile strength and the lowest water absorption compared to the rest of the mixtures, the two of them were taken forward for thermal and dimensional analysis.

### 3.5. Thermal Analysis

#### 3.5.1. Thermomechanical Analysis (TMA)

Thermal expansion of A40, A40W50, A40W50-PC, A40W50-B, and A40W50-B-PC were measured using a TMA device to assess their performance for the intended application, where minimum thermal expansion is required during the burnout process of sacrificial casting patterns. All specimens were tested from room temperature to 160 °C. To protect the device from the excessive wax sweating of the samples, no higher temperature range was measured, except for A40 which was heated up to 200 °C. The results of the analysis show the maximum amount of expansion (percentage of dimension change) in the wax filled samples (irrespective of post curing state and pigment content) is almost half that of the unfilled sample Table 5. The crosslink density of the thermoset polymers is responsible for the thermal expansion in thermosets [30]. As the presence of wax filler in the polymer matrix reduces the crosslink density of the matrix per unit area, the wax filled samples show lower thermal expansion compared to unfilled test part (Figure 5). While A40 (the unfilled sample) continues to expand throughout the heating ramp up to 200 °C, the expansion of the wax filled samples reaches their maximum when the temperature is around 150 °C. After this temperature the dimensional changes of the samples either plateaus or reduces as Figure 5 indicates.

#### 3.5.2. Thermogravimetric Analysis

Thermal degradation of A40, A40W50, and A40W50-PC samples was investigated using TGA. The results show all test samples have a residual ash content of around 1.5%. it should be noted that this is not final ash content of the mixtures as for a complete and clean burnout, a minimum dwell time of 3 to 4 h is required. The Derivative Thermogravimetry (DTG) curves shows small peaks around 150 °C in A40 and A40W50 that can be attributed to the degradation of the unreacted monomers and low molecular weight species present in the system (Figure 6). The subtle peak at 250 °C in the wax filled samples, A40W50 and A40W50-PC, is related to the thermal decomposition of the low molecular weight species of the wax filler. It can be seen in Figure 6 that the major decomposition of the wax filled samples occurs in two stages between 400 °C and 500 °C, showing two distinct peaks at 420 °C and 475 °C. In contrast with wax filled samples, the decomposition of the unfilled test sample (A40) starts at a lower temperature of just over 300 °C and continues to 500 °C, with a single peak at 450 °C. When it comes to the burnout of the casting patterns, the wider distribution of decomposition is preferred since it can potentially result in a slower gas formation during the burnout process, and hence a smaller risk of ceramic shell failure.

One of the aspirations of using filler material in the photoresin mixture was to distribute the degradation and burnout process over a larger temperature range. The DTG curves in Figure 6 reveal that adding wax filler to the mixture splits the major weight loss of the material into two stages. However, the thermal degradation of the wax filled samples starts at higher temperatures compared to that of unfilled sample, A40 (starting at 400 °C as opposed to 300 °C). The addition of wax has shortened the temperature range over which the major decomposition of samples happens. This is contradictory to our initial intentions of using wax filler in photoresin mixtures for a smoother burnout process. This observation can be explained by the latent heat of evaporation and the decomposition of lower molecular weight species of the wax filler. This means the wax particles in the photopolymer matrix consume a portion of the system’s thermal energy towards their phase transition and decomposition. This is an additional transition stage to the burnout process of the wax filled samples compared to unfilled ones. In this case only when this transition is over the degradation of the photopolymer matrix begins, hence the delayed degradation of the wax filled samples.

#### 3.5.3. Deflection Tests

Figure 7 displays the buckling behavior of A40, A40W50, and A40W50-PC samples with three thicknesses of 0.8 mm, 1.3 mm, and 1.8 mm at 100 °C, 150 °C, and 200 °C. As expected, increasing the thickness of the samples is one of the main factors inhibiting the buckling or deflection of the test strips at high temperatures. Post curing is the next important factor to contain the deflection of the samples. Regardless of the thickness and temperature, the unfilled green samples and the post-cured wax filled test strips showed maximum and minimum deflections, respectively. In line with the observations in TGA and DTG curves (Figure 6), the physical indications of degradation, e.g., flaking and cracks, are significantly more visual on the unfilled samples, A40, in all thicknesses and lower temperatures. In other words, the lack of wax and, therefore, its latent heat of evaporation and its decomposition in A40 samples has enhanced the heat absorption by the photopolymer network and consequently caused more thermal decomposition. In the wax-filled mixtures, however, the polymer matrix remains more rigid at higher temperatures, where sweating the wax out of polymer network opens some room for free thermal expansion of the polymer network without causing too many cracks and deflection.

### 3.6. Dimensional Studies

Since filled photoresins have been widely commercialized as castable resins for VPP of casting patterns, A40W50 was used to study the dimensional behavior of the prints. A medium sized blade geometry (Appendix A) was selected to investigate the dimensional accuracy and stability of this wax filled photoresin (A40W50). The impact of post curing process on dimensional behavior of A40W50-PC was also studies. Two blades using A40W50 photoresin were printed. One of the blades was post-cured (A40W50-PC) after the first measurement while the other was studied in its green (not post-cured) condition. Both blades were stored in the ambient environment and remeasured every 7 days for a 21-day period. Appendix A summarize the dimensional deviations of two cross-sections of the blades using A40W50 mixture in green (A40W50) and post-cured (A40W50-PC) conditions, respectively.

After the initial shrinkage caused by post curing, the post-cured blade, A40W50-PC, sustained more uniform dimensions and generally smaller standard deviations compared to green counterpart, A40W50, over the course of 14 days (Appendix A). During the 21-day storage period, the dimensional deviations of both blades was contained to an average of ±130 µm, which often well satisfies the requirements for the as-cast dimensions of the aero and gas turbine hot-section components. Figure 8 shows the heat map generated by comparing the 3D scanned part and its CAD model.

### 3.7. Burnout of Casting Patterns

The blade shown in Appendix A was printed using the A40W50 mixture. The burnout regime was designed according to the thermal analysis data (TGA and DGA). Therefore, the furnace was set to reach to 450 °C (in 15 min), at which approximately 45% of the network’s mass will be decomposed. A dwelling of 60 min was applied for a complete decomposition at this temperature. Afterwards in another heating ramp the furnace temperature was increased to 800 °C in 15 min to finish the burnout process. Figure 9 illustrates the ceramic shell before and after the burnout process and shows no damages to the ceramic shell and the shell, ideal for the intended application.

## 4. Conclusions

This study, for the first time, discusses the use of thermoplastic photoresins for a castable VPP resin mixture, to enhance burnability of the casting patterns and increase the chance of successful burnout of complex geometry ceramic shells with no cracks/damages, thanks to low thermal expansion and reduced high-temperature-strength of the resin mixture. The proposed mixture also offers high dimensional accuracy and stability needed for the casting patterns of aero and industrial turbine components. This solution broadens the horizon of using accessible, scalable, and cost-efficient VPP method in a wide range of casting applications for industrial and engineering products, hence shortening the supply chain and empowering design flexibilities.

Use of a monofunctional photocurable monomer as a “thermoplastic” monomer that softens upon heating, namely ACMO, for 3D printing of investment casting sacrificial patterns was investigated in this study, and the following conclusions were made:-Due to the high levels of water absorption of ACMO, HDDA was introduced to the mixture in different ratios. Increasing the HDDA content from 20 wt% to 60 wt% reduced the water absorption by almost 90%.-A sudden drop in strength of the printed material was observed at higher temperatures (above 150 °C), using deflection test, which is highly desirable in the intended application. As a result, burnability of the parts significantly improved and the risk of ceramic shell failure during burnout process was minimized. This was confirmed by the successful burnout of a turbine blade casting pattern with no shell cracking and failure.-An excellent dimensional accuracy of the VPP-printed components was achieved, using a turbine blade verification sample. Dimensional stability of the parts was also proved within the acceptable range, when stored and remeasured within 14 days of production. The dimensional deviations of the parts did not exceed an average of ±130 µm at any point during a 21-day period. The post-cured samples retained lower dimensional deviations throughout the measurement intervals.

## Figures and Tables

**Figure 1 polymers-14-04593-f001:**
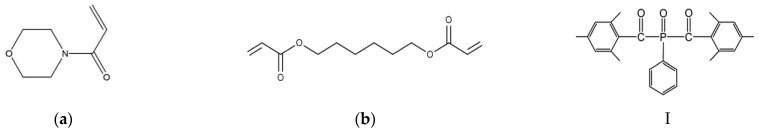
The chemical structures of the components used in this work. (**a**) Acryloyl morpholine (ACMO), (**b**) Hexanediol diacrylate (HDDA), and (**c**) Phenylbis (2,4,6-trimethylbenzoyl) phosphine oxide (BAPO).

**Figure 2 polymers-14-04593-f002:**
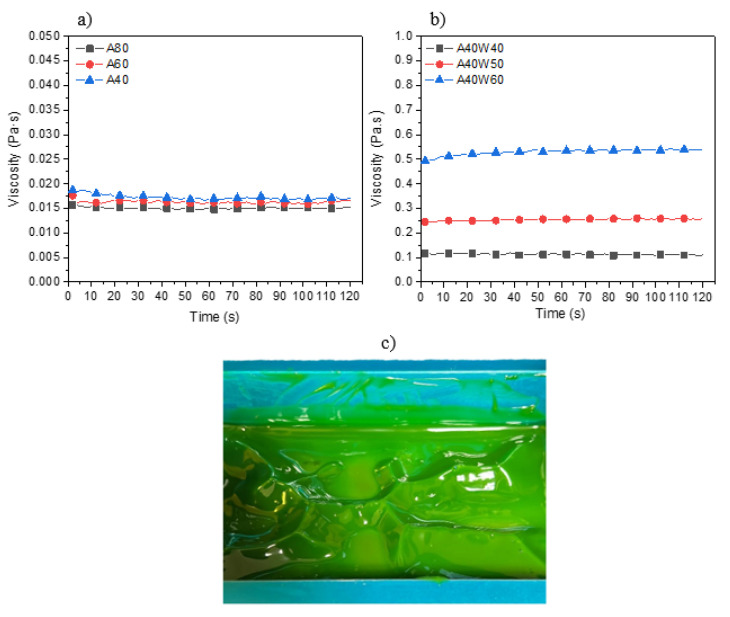
Viscosity measurements of (**a**) Unfilled and (**b**) Wax filled photoresins under constant shear rates of 500 s^−1^ and 100 s^−1^, respectively, and (**c**) the gel-like behavior of A40W60 leading to poor flowability.

**Figure 3 polymers-14-04593-f003:**
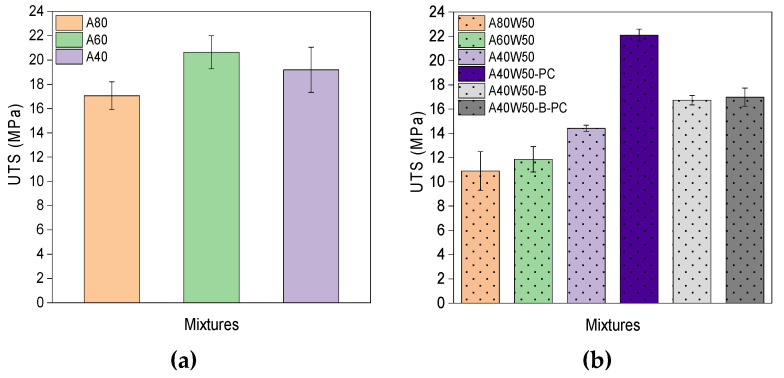
Ultimate tensile strength of (**a**) unfilled samples, and (**b**) filled samples.

**Figure 4 polymers-14-04593-f004:**
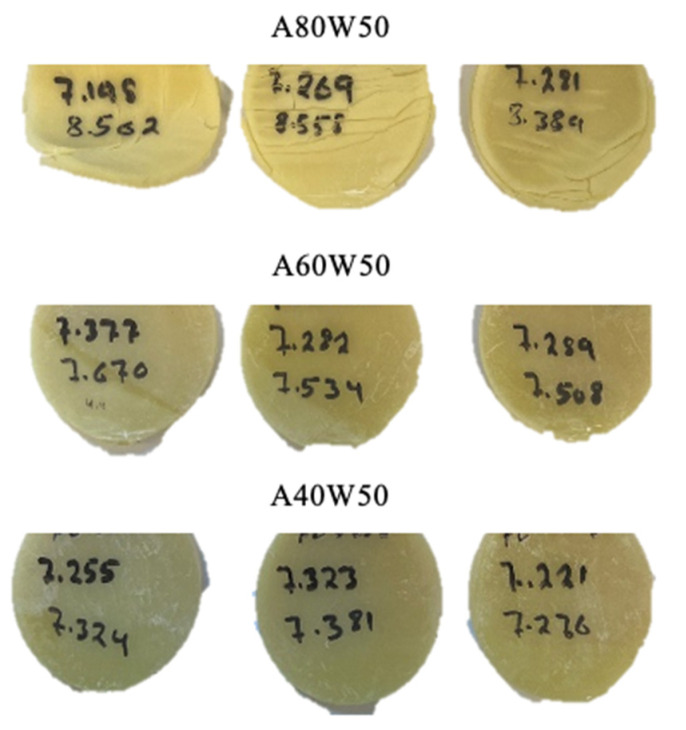
Water absorption disks after 24 h immersion in distilled water. Samples with higher content of ACMO (A80W50) showed severe deformation, swelling, and cracks due to excessive water absorption.

**Figure 5 polymers-14-04593-f005:**
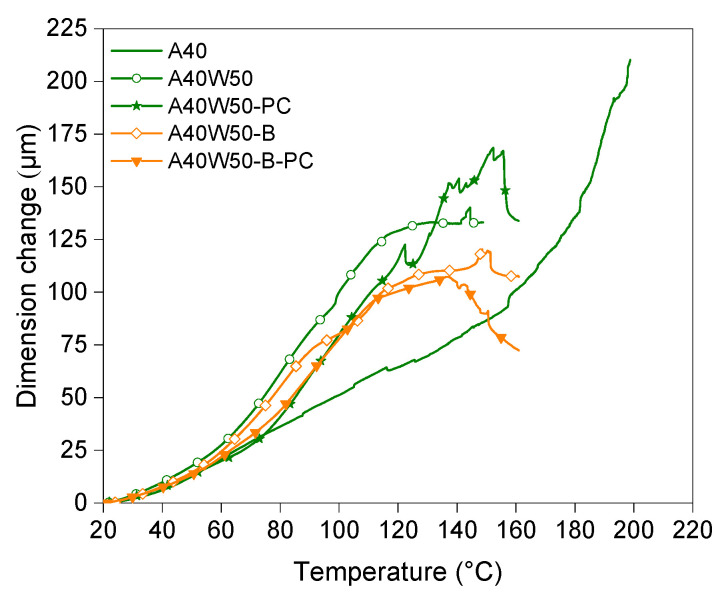
Thermogravimetric Analysis (TMA) of test specimens.

**Figure 6 polymers-14-04593-f006:**
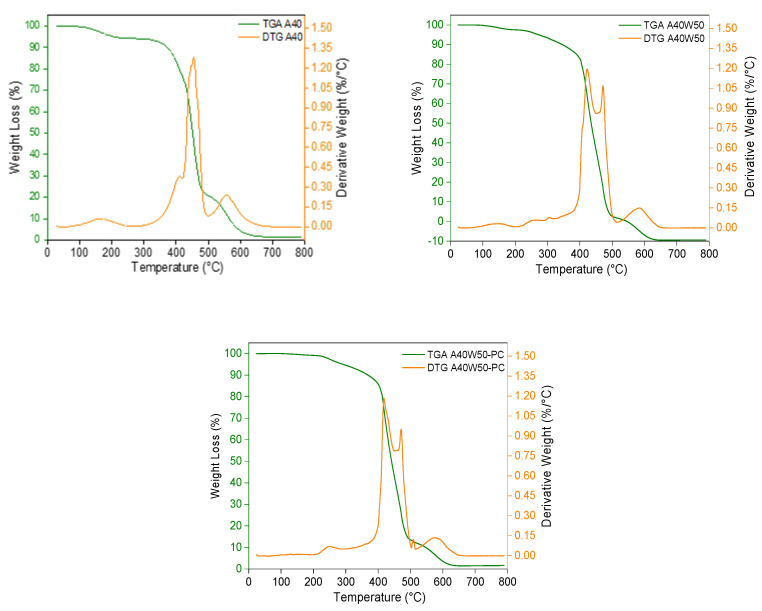
Thermogravimetric (TGA) and derivative gravimetric (DGA) analysis of three test samples of A40, A40W50, and A40W50-PC.

**Figure 7 polymers-14-04593-f007:**
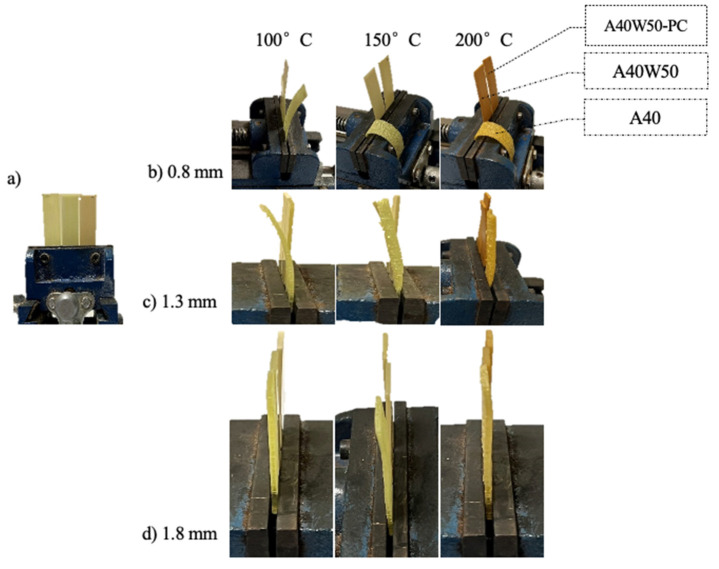
Setting of the test strips for buckling test, (**a**) before going into oven and (**b**–**d**) buckling of test strips with different thicknesses in different temperatures.

**Figure 8 polymers-14-04593-f008:**
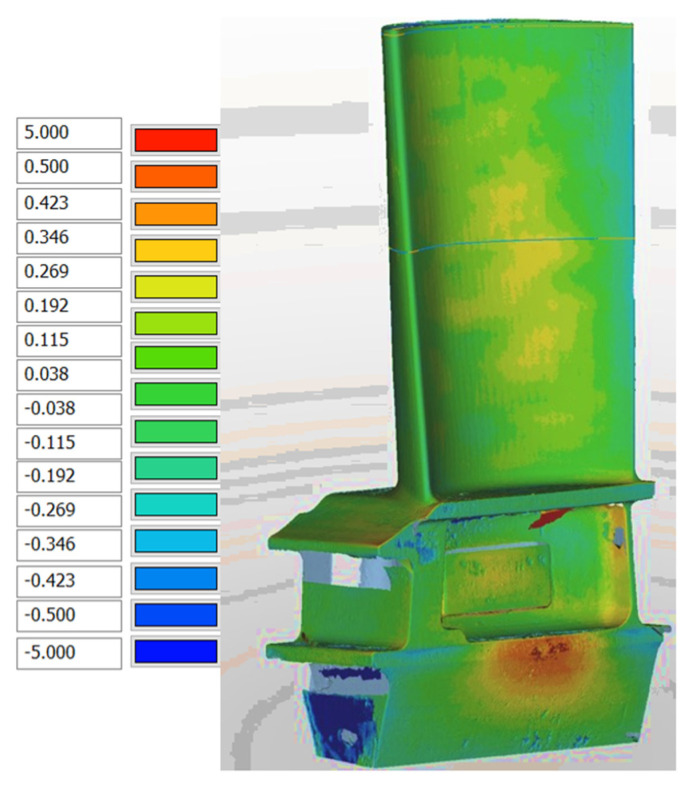
An example of the “global CAD compare” of the parts measured by Nikon 3D scanner machine and processed by Focus Inspection software.

**Figure 9 polymers-14-04593-f009:**
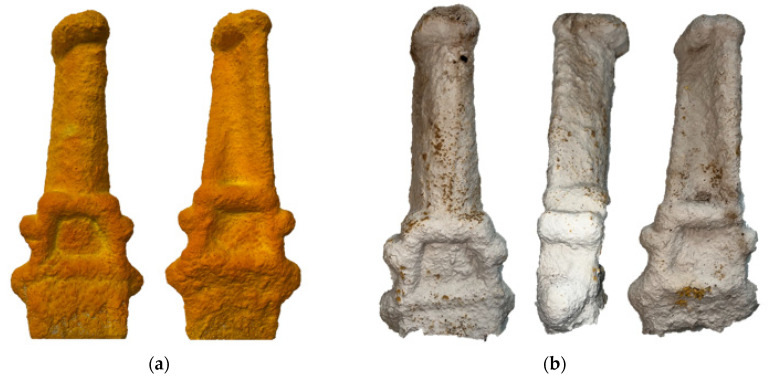
The ceramic shell before (**a**) and after (**b**) the pattern burnout process.

**Table 1 polymers-14-04593-t001:** Labelling and contents of the mixtures.

	Content	ACMO	HDDA	Wax (Vol%)	Black Pigment (wt%)	Post-Cured
Label						
A100		100	-	-	-	No
A80		80	20	-	-	No
A60		60	40	-	-	No
A40		40	60	-	-	No
A80W50		80	20	50	-	No
A60W50		60	40	50	-	No
A40W50		40	60	50	-	No
A40W40		40	60	40	-	No
A40W60		40	60	60	-	No
A40W50-B		40	60	50	1.5	No
A40W50-PC		40	60	50	-	Yes
A40W50-B-PC		40	60	50	1.5	Yes

**Table 2 polymers-14-04593-t002:** Degree of Conversion of filled and unfilled samples.

	A80	A60	A40	A80W50	A60W50	A40W50	A40W50-PC	A40W50-B	A40W50-B-PC
DC (%)	61.62	46.99	44.96	45.63	36.98	38.46	44.99	45.26	49.36

**Table 3 polymers-14-04593-t003:** Tensile strength of samples prepared using different mixtures.

	A80	A60	A40	A80W50	A60W50	A40W50	A40W50-PC	A40W50-B	A40W50-B-PC
UTS (MPa)	17.06	20.64	19.19	10.89	11.86	14.41	22.1	16.73	16.99

**Table 4 polymers-14-04593-t004:** Water absorption of samples prepared using different mixtures.

	A80	A60	A40	A80W50	A60W50	A40W50	A40W50-PC	A40W50-B	A40W50-B-PC
Water absorption (%)	17.63	5.56	1.97	17.3	3.48	0.83	0.77	0.84	0.77

**Table 5 polymers-14-04593-t005:** Thermomechanical data (CTE and dimensional change) of test samples.

	* CTE (Ambient to 160 °C) 10^−6^ k^−1^	CTE (160 to 200 °C) 10^−6^ k^−1^	** Dimension Change before/at 160 °C %	*** Dimension Change at 200 °C
A40	218.8	813.8	3	6.2
A40W50	219.4	-	2.7	-
A40W50PC	257	-	3.43	-
A40W50-B	222.1	-	2.42	-
A40W50-B-PC	226.2	-	2.2	-

* Coefficient of Thermal Expansion. ** This is the maximum dimension change for wax filled samples. *** This is the maximum dimension change of unfilled sample.

## Data Availability

Data are available on request due to restrictions. The data presented in this study are available on request from the corresponding author.

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
