# Peer review of "Large Scale Vat-Photopolymerization of Investment Casting Master Patterns: The Total Solution"

_polymers, 2022, doi:10.3390/polym14214593_

Round 1
Reviewer 1 Report
Present manuscript deals with the study of large scale vat-photopolymerization of investment casting master patterns. It seems to be interesting and it can be accepted after minor revision.
Comments,
1) In introduction, recent works should be discussed and cited.
2) Novelty of the work is not clear.
3) 3.5. Thermal analysis should be discussed in detail.
4) Conclusions should be rewritten.
5) Reference sections should be updated.
Author Response
Present manuscript deals with the study of large-scale vat-photopolymerization of investment casting master patterns. It seems to be interesting, and it can be accepted after minor revision.
Thank you very much for your feedback and interest in this manuscript. The comments have been addressed as below:
Comments,
1) In introduction, recent works should be discussed and cited.
Thank you for your comment. We can now confirm that some further recent works that we could find and related to the subject have been added to the introduction section in page 3. Please advise if we miss any prominent work in the field.
2) Novelty of the work is not clear.
We improved the last paragraph of the introduction to emphasise the novelty of this work.
Briefly, on one hand, the current literature is lacking a wholesome solution regarding the successful implication of additively manufactured casting patterns of highly precise and accurate hot section components for gas and aero turbine components. On the other hand, using photocurable monomers to 3D print thermoplastic-like parts is a novel area with only a few existing research articles available in public domain. In this manuscript we are presenting a complete solution to 3D print casting patterns with high precision and successful burnout performance.
3) Thermal analysis should be discussed in detail.
This section has now improved in the revised manuscript. Please see section 3.5.
4) Conclusions should be rewritten.
Thank you for your comment. The conclusion section has now been rewritten and restructured for a better read and flow, as advised.
5) Reference sections should be updated.
The reference section is now corrected and updated accordingly.
Reviewer 2 Report
This work mainly used two kinds of monomers, i.e. ACMO and HDDA for 3D printing of casting patterns, and the effects of wax filler was also studied. The results presented in this work are very simple and the originality is low. The framework of this manuscript needs improved and the authors need a better writing skill. After checking this manuscript or draft, I can’t recommend publication in POLYMERS for the moment. Actually this draft needs total improvement from some experts, who have already published some papers. My opinions were listed below:
(1). In the last paragraph of the “Introduction” part, I believe some sentences were directly copied somewhere. This is a very big mistake.
(2). The chemical structures of monomers (ACMO and HDDA) and photoinitiator (BAPO) should be added into the manuscript as new Scheme 1, so that the readers will have a better understanding.
(3). Figure 1 can be transferred into the “Supplementary” part.
(4). The caption of Table 2 is “Degree of Conversion, Ultimate tensile strength, and Water absorption of filled and unfilled samples.” However, only Degree of Conversion (DC) values were presented in Table 2. Why?
(5). The format of the “Reference” part is not consistent, such as the author list of Ref. 8, Ref. 33, Ref. 34 and Ref. 36.
(6). At the end of the manuscript, Table S1 should be deleted, since it already appeared in the “Supplementary” part.
Author Response
We would like to thank the reviewer for their detailed reviews and feedback. The comments have been addressed as below:
My opinions were listed below:
(1). In the last paragraph of the “Introduction” part, I believe some sentences were directly copied somewhere. This is a very big mistake.
We wish to thank the reviewer for this comment and acknowledge our shortsighted mistake. It’s worth nothing to explain that this paragraph was left from the Journal’s “template” to restructure the paper for the submission, and hence the sentences the reviewer rightly pointed out have remained from the original template. This section has now been removed from the manuscript.
(2). The chemical structures of monomers (ACMO and HDDA) and photoinitiator (BAPO) should be added into the manuscript as new Scheme 1, so that the readers will have a better understanding.
Thank you for your comment, this suggestion certainly helps the readers with a better understanding. The chemical structures of the monomers and photoinitiator are now presented as Figure 1
(3). Figure 1 can be transferred into the “Supplementary” part.
This figure has now been moved to supplementary documents.
(4). The caption of Table 2 is “Degree of Conversion, Ultimate tensile strength, and Water absorption of filled and unfilled samples.” However, only Degree of Conversion (DC) values were presented in Table 2. Why?
Thanks for rightly pointing this out. The caption is now corrected. In the initial draft of this manuscript tables 2, 3, and 4 were merged in one single table (as table 2), we decided to separate and place them in their relevant sections in latest revision. The caption unfortunately was left unchanged.
(5). The format of the “Reference” part is not consistent, such as the author list of Ref. 8, Ref. 33, Ref. 34 and Ref. 36.
The reference section and in particular the abovementioned references are now corrected in the reference list.
(6). At the end of the manuscript, Table S1 should be deleted since it already appeared in the “Supplementary” part.
Table S1 is now removed from the manuscript.
Reviewer 3 Report
Many sentences are hard to read. Correct English in detail.
Author Response
We would like to thank the reviewer for their feedback. We confirm that we reviewed and corrected the writing style throughout the manuscript for a better read.
Reviewer 4 Report
The engine blade has a complex and fine geometric structure, and requires very high dimensional accuracy. Therefore, it will be produced by precision casting, also known as wax loss casting, which makes the manufacturing of high-precision wax mold a key step. The rise of UV curing 3D printing technology can produce products with complex structures, but it poses new challenges to the material ratio. In this paper, the ratio, rheological behavior, mechanical properties, burning conversion and dimensional accuracy of UV curable materials were systematically studied. Many optimized material parameters were obtained, and good UV curable wax models were obtained. The research method of the paper is reasonable, and the research results have important application value and can be published.
Author Response
We wish to thank the reviewer for their positive feedback and interest in our manuscript. This is very much appreciated.
Round 2
Reviewer 2 Report
The authors have addressed some of my questions.
But it's better to leave it to the editor to make the decision.